# Communication in Dogs

**DOI:** 10.3390/ani8080131

**Published:** 2018-07-31

**Authors:** Marcello Siniscalchi, Serenella d’Ingeo, Michele Minunno, Angelo Quaranta

**Affiliations:** Department of Veterinary Medicine, Section of Behavioral Sciences and Animal Bioethics, University of Bari “Aldo Moro”, 70121 Bari, Italy; serenella.dingeo@uniba.it (S.d.); m.minunno79@libero.it (M.M.); angelo.quaranta@uniba.it (A.Q.)

**Keywords:** dog, communication, behaviour

## Abstract

**Simple Summary:**

Communication takes place between members of the same species, as well as between heterospecific individuals, such as the long co-habitation process and inter-dependent relationship present in domestic dogs and humans. Dogs engage in visual communication by modifying different parts of their body; in tactile communication; and also in auditory and olfactory communication, with vocalizations and body odours, respectively. The aim of this review is to provide an overview of the recent literature about dog communication, describing the different nature of the signals used in conspecific and heterospecific interactions and their communicative meaning. Lateralized dog brain patterns underlying basic neural mechanisms are also discussed, for both conspecific and heterospecific social communication.

**Abstract:**

Dogs have a vast and flexible repertoire of visual, acoustic, and olfactory signals that allow an expressive and fine tuned conspecific and dog–human communication. Dogs use this behavioural repertoire when communicating with humans, employing the same signals used during conspecific interactions, some of which can acquire and carry a different meaning when directed toward humans. The aim of this review is to provide an overview of the latest progress made in the study of dog communication, describing the different nature of the signals used in conspecific (dog–dog) and heterospecific (dog–human) interactions and their communicative meaning. Finally, behavioural asymmetries that reflect lateralized neural patterns involved in both dog–dog and dog–human social communication are discussed.

## 1. Introduction

Communication occurs between members of the same species, as well as between heterospecific individuals, as occurs between domestic dogs and humans [1]. Living in close contact with humans for at least 30,000 years [2], dogs have developed specific skills enabling them to communicate flexibly with humans [3]. There is now evidence suggesting that the dog–human relationship can be characterized as an “attachment”, which closely resembles the one reported between infants and their primary caregivers [4,5]. Specifically, the co-habitation process and the human–dog attachment caused both in human and in dogs changes in their cross-species communicative abilities, the result of which is to perceive and understand the other species’ signals and correctly respond to them [6].

Dogs show a flexible behavioural repertoire when communicating with humans, employing the same signals used in intraspecific interactions (dog–dog), some of which can acquire and carry a different meaning when used toward humans (e.g., eye contact, [7]). They use their whole body to communicate, conveying information intentionally or otherwise [8]. Not all the signals, in fact, are under voluntary control. When a dog experiences an emotional state, for example anxiety, it releases a specific body odour into the environment [8,9]. Despite being involuntary, this signal is received as a communicative signal by other individual because it informs them about the sender’s inner state and it can produce changes in the receiver’s behaviour [8]. 

Dogs are engaged in visual communication by modifying different parts of their body, in tactile communication, and in auditory and olfactory communication, with vocalizations and body odours, respectively. The aim of this review is to provide an overview on the recent literature about dog communication, describing the different nature of the signals used in conspecific and heterospecific interactions and their communicative meaning.

## 2. Visual Communication

Dogs communicate visually with other individual modifying the position of different parts of their body (see Figure 1 and Figure 2). Control by voluntary muscles allows dogs to display a wide range of postures and body part positions that convey different information about the signaler’s inner state and intentions [8]. However, humans, through artificial selection over many years, have produced changes in dogs’ anatomy and morphology that have reduced the social signaling capacity of several breeds [10]. For instance, brachycephalic dog lost the flexibility in displaying different facial expressions and dogs with permanently erected ears or with a very short tails lost part of their behavioural repertoire expressed by these anatomical structures [10]. The long or dense fur of some breeds obscures several visual signals, like piloerection, or even entire parts of dogs’ body (eyes, mouth, or legs) [10,11]. Therefore, visual communication could be extremely challenging for some dogs, both for correctly delivering and for interpreting visual information.

Broadly speaking, individuals’ proximity and direct interactions are required during visual communication [12]. In dogs’ encounters with other conspecifics, body size and body posture are the first visual signals perceived, providing the very first information about other individuals’ intentions [10]. Dogs can communicate confidence, alertness, or threat by increasing their body size, pulling themselves up to their full height, and increasing the tension of the body muscles [8]. The individual’s body size can be further increased by piloerection of the hackles (Figure 1). The piloerection reflex occurs in several contexts related to the individual’s increase in arousal, indicating, for example, fear or surprise, or to communicate aggression or stress [8]. It still remains a question to be further investigated whether the location of the raised hackles on the dog’s body could be really informative about dogs’ emotional states.

On the other hand, dogs can reduce sizes perceived by other individuals by lowering their body and their tail and flattening back their ears to avoid conflicts or during stressful interactions [8,11] (Figure 3).

The tail contributes to help define postural displays and its positions and movements are used to convey different information about the individuals’ emotional state and intentions. The tail is held high to communicate confidence, arousal, or the dog’s willingness to positively approach another individual, for example greeting and playing [10,11], while it is held stiff to express a threat or the individual’s anxiety [8,10,11]. On the contrary, a tail held low or tucked between the limbs signals fear, anxiety, or appeasement as it contributes to decreasing the individual’s body size [8,10,11]. Dogs wag their tails loosely from side to side to communicate friendliness or their excitability [8]. Fast movements of the tail, instead, express different inner states according to its position; dogs communicate confidence if they hold their tail high, while a low wagging is generally associated with anxiousness, nervousness, or internal conflict [8,10]. There is now evidence that the direction of tail wagging movements is also directly involved in intraspecific communication. Specifically, when dogs look at stimuli with a positive emotional valence (e.g., their owner), their tail moves more towards the right side; on the other hand, when dogs look at clear negative emotional stimuli (an unfamiliar dog with a clear agonistic behaviour), a higher amplitude of tail wagging to the left appears. Given that the movement of the tail depends on the contralateral side of the brain [13], left–right directions of tail wagging are consistent with Davidson’s laterality–valence hypothesis about the specialization of the left side of the brain for the control of approaching behavioural responses (right-wag for positive stimulus) and the main role of the right side of the brain for the control of withdrawal responses (left-wag for negative stimulus) [14]. The decisive aspect for visual intraspecific communication is that dogs seem to be able to detect tail movement asymmetries of other conspecifics, and thus indirectly deduce their emotional state [15].

In close-range social interactions, dogs can also obtain and deliver information about their inner state through their facial expression, modifying gaze, ears, and mouth position (Figure 3 and Figure 4). Previously, facial expressions were considered involuntary displays of an individual’s emotional state. However, recent research has discovered that dogs produce facial expressions as an active attempt to communicate with others [16].

The eye region plays an important informative role in face recognition in dogs. Dogs usually stare at other individuals to threaten them, while they avoid making eye contact to appease and to decrease the tension during an interaction [8,10]. Eye tracking studies demonstrate, indeed, that dogs address their attention principally to the eye region when processing conspecific faces [17,18]. Canine eyes can communicate individuals’ emotional states. Eyes are “soft” in relaxing and non-threatening contexts (Figure 4 and Figure 5), while they are “hard” when partially open and with brow wrinkled, expressing a high level of tension (Figure 6) [8].

In agonistic and stressful situations, dogs can open their eyes wide, exposing the whites of the eyes, namely the sclera (“whale eyes”) [8]. Moreover, dogs can derive information about other individuals’ intentions by evaluating their willingness to make eye contact, especially in agonistic contexts. Coloured markings around the eyes (e.g., small brown spots above the eyebrow ridge of Dobermans and Rottweillers), could favour attention catching toward the eye region in order to facilitate the interpretation of conspecific communicative signals conveyed by different facial expressions [8]. This hypothesis is supported by the fact that there is now clear scientific evidence that, in dogs, colour information may be predominant with respect to brightness [19,20].

Along with the eyes, ear position represents a relevant signal in individuals’ emotional expression, even though its role in face processing has been rarely investigated. As highlighted above for the tail, it is necessary to consider breed differences in the morphology of the ears and ability to move them when defining the “relaxed” position, and the different changes should be evaluated by examining any ear base modifications [11]. Generally speaking, dogs can pull their ears back various degrees according to the animals’ arousal state. Ears can vary from simply “back”, to communicate an appeasement intention, to “flattened” or “pressed back”, in frightened individuals (Figure 3) or as an agonistic response (Figure 6). In extremely fearful individuals, ears can be pressed back so far on the head that they completely disappear (“seal ears”) [8]. On the contrary, ears kept forward are associated with interest, attention, and approach-oriented intentions [11], while sideward position indicates a conflicting inner state (“airplane ears”) [8].

Although the mouth region is less investigated compared with the eyes when dogs process conspecific faces, the mouth acquires a particular importance when evaluating whether a facial expression is potentially threatening. Dogs, indeed, look more at the mouth region of pictures portraying threatening and neutral conspecific facial expressions [18]. In our opinion, the eyes staring at a fixed point that is displayed in the “neutral” expression could be interpreted as “eye stalking” by the receiver, who can focus their attention on the mouth to perceive other information (lip position or a vocalization) to correctly interpret this expression.

Mouth configuration varies according to its position (open or close) and to the labial commissures shape, which conveys important information about the individual’s aggressive intentions and its stress state. The labial commissure of the mouth is drawn forward (“short lips”) in agonistic displays (Figure 6) and the related opening degree of the mouth increases according to threat intensity [8]. On the contrary, dogs pull back their labial commissure (“long lips”) to communicate stress [8,11], the intensity of which increases if the commissures are drawn more backward and form a “C” shape [8]. 

Along with postural and facial displays, dogs can exhibit other behaviours to signal their inner state; for example, they turn their head away from a stimulus when stressed (Figure 3B), they lift their forehead paw to indicate uncertainty, or they lick their lips to communicate their appeasement intentions [8,21].

Overall, despite the fact that communication behaviours can be described separately, single behaviours need to be considered and observed in the context of all the other signals displayed at that time, as well as the general body language, in order to interpret correctly the individual’s emotional state.

Dog–human communication has received growing interest over the past twenty years. In particular, several studies investigating dogs’ comprehension of human visual signals revealed that dogs are tuned into human visual communication [3]. Dogs, indeed, already show a high sensitivity to human-given cues in an early stage of their development [22,23,24], following spontaneously human body postures, gaze direction, and pointing to find a target location [25,26,27]. They also prefer to rely more on human gestures rather than auditory cues in a two-choice task, in which the information received is contradictory, suggesting that gestures are more salient for them [28].

Most importantly, recent studies reported that dogs are skilful in interpreting the communicative intent of humans by understanding the ostensive-referential nature of specific signals, such as eye contact or directed-speech [29,30,31]. Ostensive cues are a characteristic element of human communicative interactions that express the sender’s intention to initiate a communicative interaction [7]. Thus, dogs’ ability to recognise human ostensive signals, which is unique in the animal kingdom, suggests a high level of adaptation to the human social environment [30]. Furthermore, the flexible comprehension of human gestures allows dogs to efficiently discriminate which of the numerous and different human social behaviours displayed in the everyday life are directed to them [7]. Dogs, indeed, evaluate the same behaviour differently according to the presence of an ostensive cue that precedes or accompanies it, ignoring the unintended movements [29]. Among human ostensive signals, eye contact represents the most important and efficient one [7,29]. From an early age, dogs show a spontaneous tendency to gaze at human faces and to make eye contact [32] in a wide range of contexts, for example, in unsolvable tasks or to beg for food from humans [33,34]. Given the specific nature of the contexts in which it is displayed, the human-directed gaze has been interpreted as a “request of help” [7,33]. Thus, dogs use eye contact to communicate with humans differently from conspecific communication, in which it represents a clear threatening signal [8]. On the contrary, in interspecific communication, and in a friendly context, it facilitates the beginning and the maintenance of human–dog interaction [35]. Therefore, through the domestication process, dogs have modified the functional meaning of this typical behavioural pattern to adapt it to a cross-species communication, acquiring a human-like communication mode [30]. Furthermore, human–dog mutual gaze enhances the establishment of an affiliative relationship and a social bond between dogs and humans by the same oxytocin-mediated effect described for mother–infant dyad and for human sexual partners [5,36,37].

The informative role of the eyes for human–dog communication is also demonstrated by the greater interest by canids in investigating the eye region compared with the other inner facial features in processing human faces [17]. Moreover, dogs assess human’s attentional state during communicative interactions by evaluating human gaze direction and adapting their behaviour accordingly [38]. It has been recently shown that human attention affects dogs’ facial expression production, as dogs increase all facial movements when a human is attending to them. This evidence highlights both dogs’ ability to act differently according to humans’ readiness to interact with them and, more importantly, dogs’ communicative intent in producing facial expressions. In particular, “tongue show” and “inner brow raiser” facial movements are used as flexible signals to catch human attention, because, for example, the “eyebrow raising” triggers human innate tendency to respond to this ostensive signal [16]. Dogs mainly rely on humans’ availability to make eye contact when they communicate with them, increasing their visual communicative behaviour according to their presence. In particular, eye contact has a crucial role for the dogs’ referential communication with humans [38]. Dogs, indeed, are able not only to flexibly use human gaze to regulate their behaviour in specific contexts, but also to communicate with humans to direct their attention to a specific object of their interest, by performing the so-called “showing behaviour” [33,39]. It has been recently reported that dogs can use up to 19 different referential gestures during everyday interactions with humans, eliciting humans’ appropriate responses [6]. They use their body position and sustained gaze as a local enhancement signal [40] or they alternate their gaze between the target object and humans to indicate to them the object location [33,41]. These signals are displayed to communicate with humans and are modulated both by human availability to communicate with them [42] and by human responses. Dogs, indeed, produce persistently referential signals until they elicit a satisfactory human response [43], but they are also able to interrupt them when they are no longer successful [44].

Recent studies have demonstrated the existence of behavioural synchronization between dogs and humans (see for review [45]). The canine synchronizes its locomotor behaviour with that of its owner in different contexts, both indoors [46] and outdoors [47], and when facing an unfamiliar human. Dogs synchronize their behaviour with the owner’s withdrawal response toward strangers, taking longer time to approach them [48]. It has also been reported that the behavioural synchronization phenomenon is affected by dogs’ affiliation toward humans; pet dogs show a higher performance in synchronizing their behaviour with their owner’ than shelter dogs with their caregivers. Moreover, behavioural synchronization affects dogs’ social preference toward humans, and in particular, toward individuals synchronizing their locomotor activity with them [45]. Thus, authors concluded that, as previously described in humans, this phenomenon increases social cohesion and affiliation in dog–human dyads, contributing to emotional contagion [49].

Despite dogs’ high social competence to communicate and interact with humans and to perceive and correctly respond to their signals, there are some open questions that still need to be addressed.

Although dogs react to the informative nature of human ostensive-referential cues, they may interpret human gestures as an order rather than understanding the human communicative intent to share information [3,30]. This hypothesis is supported by dogs’ higher attitude to follow owners’ signals rather than those from a stranger and to follow human gestures to locate food even if the olfactory information about its position is contradictory [1]. Furthermore, it has been reported in a recent eye-tracking study that dogs are able to discriminate between social and non-social interactions depicted on a picture, showing a longer gaze toward the individuals in a social context compared with a non-social one [50].

A further important aspect of dog–human visual communication is the ability to perceive other individuals’ emotions expressed by their faces. There is broad evidence that both dogs and humans are skilled in recognizing the other species’ emotions by looking at their faces [51,52]. Specifically, domestic dogs show a functional understanding of human emotional facial expressions, responding differently according to its valence. They regulate their behaviour toward an unknown or ambiguous object by using human emotional referential expressions, especially when provided by their owner; dogs prefer to approach it or to stay away if the human expresses happiness or fear/disgust, respectively [53,54]. Interestingly, when the informants are inattentive, dogs actively attempt to involve them to obtain information, alternating their gaze between the object and them. Taken together, these findings demonstrate the existence of social referencing in dogs [53].

Dogs’ perception of human emotions allows them to adjust their behaviour during everyday interactions with humans and to respond appropriately. A recent study reports that dogs display mouth-licking behaviour, which is a stress indicator, more often when presented with negative emotional facial expressions compared with positive ones [55]. This evidence suggests both that they perceived the negative valence of the human emotion, increasing their level of stress, and that they responded adequately to it, displaying a behaviour used in conspecific communication to “appease” the sender [8]. Recent scientific literature shows that emotional cues conveyed by human emotional faces are processed in an asymmetrical way by the canine brain. Specifically, using a behavioural method commonly employed to study both visual and auditory lateralization (namely the “head orienting paradigm” [56,57]), it has been shown that dogs are sensitive to human faces expressing Ekman’s six basic emotions (e.g., anger, fear, happiness, sadness, surprise, disgust, and neutral) with a specialization of the right hemisphere for the analysis of human faces expressing “anger”, “fear” and, “happiness” emotions, but an opposite bias (left hemisphere) for human faces expressing “surprise” [57].

## 3. Acoustic Communication

Domestic dogs have a broad and sophisticated vocal repertoire [58]. Although their vocalizations are similar to their closest relative, the wolf, dogs vocalize in a wider variety of social contexts compared with wolves and they retain this characteristic even into adulthood [59]. Dogs’ vocal behaviour underwent considerable changes during the domestication process, generally considered as a result of dogs’ adaptation to the human social environment [60]. The effect of living in proximity to humans on animals’ vocal behaviour has been demonstrated, indeed, by a pioneering study showing that, after a 40-year selection, tame red foxes emitted more human-directed vocalizations than their ancestors [61]. Thus, as described for the foxes, dogs could have acquired a tendency to vocalize more during interactions with humans, which could have been artificially selected, together with other socio-cognitive abilities of understanding human cues. Dogs developed, therefore, novel forms of the pre-existing vocalizations, which acquired different acoustic and functional characteristics, facilitating their communication with humans [59]. Humans, indeed, are able to derive information from dogs’ vocalisations, not only about the senders’ physical characteristics, rating, for example, growls produced by larger dogs as more aggressive than those of smaller dogs [62], but also about its emotional state [63,64]. The development of different and numerous vocal signals in dogs could have been modulated, therefore, by their efficacy of conveying specific information to communicate with humans. This hypothesis is further supported by the existence of an individual variability of the acoustic features of barks directed to humans in non-agonistic contexts (during ball play, in requesting situations, or before going for a walk), which can be shaped according to the owner’s response during everyday interactions [59,65].

The specific role of auditory signals in communication with humans is confirmed by the significant decrease of their production in feral and stray dogs [59], suggesting that dogs’ social contact with humans represents the main regulatory factor of their expression. 

Here, we provide an overview of dogs’ most common vocalizations, focusing on their functional–contextual features, both in intraspecific and heterospecific communication. 

Among the different vocal signals, the bark is certainly the most typical vocalization of dogs. Contrary to previous beliefs, which claimed that barks are a byproduct of domestication lacking any functional value, recent studies demonstrated the context-related acoustical features of barks [60,66,67], suggesting that they are means of communication in dogs.

Barks are short, explosive, and repetitive signals, with a highly variable acoustic structure (dominant frequency range between 160 and 2630 Hz), differing between breeds and even between individuals [60,66]. They are generally used in short-range interactions and in several behavioural contexts, like greeting, warning/alerting, calling for attention, or during play [58]. Moreover, barking is an allomimetic behaviour, that is, a group activity in which several individuals bark in unison with other conspecifics, mirroring and stimulating each other [8].

Dog breeds show a different use of barks in their vocal communication. Wolf-related breeds, for example, Shar-pei, Chow-Chow or Basenji, have a very rare propensity to bark, whereas other breeds present a specific type of barking, such as hunting dogs [59].

Barks carry various information about the signaler’s physical characteristics, familiarity, and inner state [62,67,68], allowing dogs to differentiate not only between barks produced by different individuals in the same context [68], but also between the different contexts in which they are produced [67]. Recent studies report, indeed, that the barks acoustic features vary predictably according to the context; dogs emit longer and lower frequency barks when a stranger approaches them, while high pitched barks are mainly produced in isolation situations [63,66]. Dogs distinguish between the different acoustic structure of barks and react accordingly to its content and the familiarity of the signaler, staying closer to the gate of their house in response to an unfamiliar dog barking at a stranger and remaining inside the house during the barks of a lonely familiar dog [67]. These findings demonstrate that barks have a functional role in intra-specific communication.

Recent studies have reported that, similar to barks, growls also convey meaningful information to dogs [62,64,69]. These low-frequency broadband vocalizations are mainly produced during agonistic interactions as a warning or threatening signal or during play interactions [8,58]. Canines can assess the body size of another individual by listening to its growl, correctly matching the sound heard with the picture portraying it [69]. Moreover, they discriminate between growls produced in different contexts, showing more inhibited behaviour to take a bone if a “guarding” growl is played [64]. It has recently been found that dogs’ growls have a context-dependent acoustics structure; in particular, its temporal features, fundamental frequency, and formant dispersions differ between play and aggressive growls, produced to threaten a stranger or to guard a bone [62,64]. In spite of the specific acoustic characteristics of growls produced during play with humans (short and high-pitched), these vocalizations, in particular, all “play vocalizations”, which also include barks and huffing [8], are less distinguishable for dogs compared with those recorded in disturbing and isolation situations [56]. Dogs’ difficulty in clearly perceiving these vocalizations can be due to the lack of other metasignals, for example, visual signals, that provide further contextual information, helping dogs to correctly interpret them [8]. The context specificity of growls and the different reaction of dogs to the different “context-type” growls demonstrate its important role in communication between dogs.

Dogs’ acoustic communication includes whines, which are indicators of stressful arousal but also greeting and attention-seeking behaviours [8]; howls, which maintain group cohesion; groans and yelps, signs of acute distress and acute pain, respectively; and grunts, which are considered as pleasure-related signals [58]. It has recently been found that canines can extract information about the emotional state of other dogs from their vocalizations. In fact, they can correctly identify the emotional valence of conspecific vocalizations, associating playful and aggressive ones with the corresponding emotional faces [70].

Moreover, conspecific vocalizations in the dog brain, as in other vertebrates, are analysed mainly by the left hemisphere, and its involvement depends on the characteristics of the sound. For example, when dogs were presented with the reversed temporal acoustic features of their calls (e.g., canine vocalizations of play, disturbance, and isolation), a shift from a left-hemisphere bias (normal call versions) to a right-hemisphere bias (play calls) or to no brain asymmetry (disturbance and isolation calls) has been reported. In addition, it is interesting to note that when intraspecific vocalizations elicit intense emotions, a right hemisphere bias appeared, confirming the hypothesis on the role of the right side of the dog brain in the analysis of arousing communicative signals [71].

Dogs and humans use vocal signals in cross-species communicative interactions that are able to produce changes in other species behaviours [72].

On one hand, canines understand the meaning of some human words and perceive the emotional content of human vocalizations. They are able to learn up to 200 words’ meaning and they link it with the object they refer to [73]. Furthermore, dogs use human voice intonation as a social referential cue, extracting information about people’s reaction to novel or ambiguous objects and acting accordingly [53,54,74]. Although vocal signals are less significant than visual ones in guiding dogs in ambiguous choice situations [43,74], the tone of human voice seems to be more efficient in communicating the human’s motive [75]. A recent study shows that dogs regulate their behaviour according to the humans’ intentions expressed by different verbal utterances, following human gestures when they are given with a cooperative intention, but ignoring them if given with a low-pitched/prohibiting voice [75]. Moreover, canines detect the intention of humans to engage in playing interactions when the human’s postural signals are accompanied by vocalizations, suggesting a specific play-eliciting function of vocalizations in human–dog social interaction [76].

The ability of the dog to correctly interpret the emotional valence of a sound also extends to human vocalizations. Specifically, using a cross-modal paradigm, it has been demonstrated that dogs can correctly match “happy” or “angry” human faces with a vocalization expressing the same emotional valence [70]. Furthermore, recent research indicates that human emotional vocalizations are processed in an asymmetrical way by the dog brain, with the prevalent use of the right hemisphere in the analysis of vocalizations with a clear negative emotional valence (i.e., “fear” and “sadness”) and the main use of the left hemisphere in the analysis of positive vocalization (“happiness”) [77].

On the other hand, dogs use vocalizations to communicate with humans, particularly to solicit their care and to attract their attention when faced with an unsolvable problem [10,33]. They mainly use short-distance calls in interactions with humans, like barks, growls, and whines, compared with long distance calls, which are used instead to communicate with conspecifics [60]. Despite little information about the intentional control of vocal productions [78], a growing body of literature demonstrates that dogs’ vocalizations are effective means for interspecific communication, conveying information for humans [63,79,80]. Humans are able to assess the signaler’s size by listening to its growls [81] and, more interestingly, they perceive the emotional content and attribute contexts to different dogs’ vocalizations [62,63,79,80]. They generally interpret growls from large dogs as being more aggressive than those emitted by smaller dogs [82], and they correctly attribute emotions to growls according to their social context (play, threatening, or food guarding) [80].

Humans can also categorize barks according to their emotional content, rating barks directed toward a stranger as more aggressive, barks produced in an isolation situation as more “despaired”, and barks recorded during play interactions as happier [63]. Moreover, regarding the acoustical structure of barks, humans generally rate low-frequency, low tonality, rapid-pulsating barks as more aggressive, while more tonal, high pitched, and slow-pulsating barks are considered to be happier or more desperate [63]. Recent findings demonstrate that humans rely on the same acoustical rules to assess the emotional content and the context of dogs’ and conspecific vocalizations, suggesting the existence of a wider common mechanism of animals to express emotions through vocal signals and to apply the same rules to encode other individuals’ inner states by listening to their vocalizations [83]. Moreover, the humans’ ability to categorize dogs’ vocal signals is independent from previous experience with dogs because the same performance has been found in recognizing dogs vocalizations in adults and five-year-old children, as well as in congenitally blind people [63,84,85].

Overall, humans’ ability to categorize dogs’ vocalizations demonstrates that dogs’ vocal signals have a communicative relevance for humans and that they represent effective means for dog–human communication.

## 4. Olfactory Communication

There are relatively few studies about the role of olfaction in dogs’ communication with both conspecifics and humans. The little research attention to the chemosignals characteristics of conveying senders’ information might be due to the minor role of olfaction in human–human communication (compared with vision and hearing) and to human minor sensibility to odours [10]. However, dogs’ high olfactory sensitivity (10,000–100,000 times higher than humans’) allows them to access social and contextual information through their sense of smell [11,12]. Body odours contain chemical signals that have specifically evolved to communicate with other individuals (Figure 7) [86]. Nevertheless, to date, dogs’ perception of the different information conveyed by odours and their role in social interaction are scarcely investigated and future studies are needed to address this issue.

Broadly speaking, olfactory communication is extremely efficient as odours persist in the environment, allowing animals to acquire information of the signaler without requiring the individuals’ physical proximity and direct interactions [12]. The olfactory signal release is below the threshold of consciousness [87] and it can occur without a communicative intent, as described for individual, gender, and emotion-related information [9,88]. Nevertheless, dogs can intentionally deposit their odour in the environment (“mark”) through urines, faeces, and glandular secretions; this behaviour is known as scent marking (Figure 7). Therefore, olfactory communication in dogs takes place via a direct interaction between two individuals through close olfactory inspection, but it also occurs indirectly through scent marking [8].

Regarding dogs’ perception of conspecific and human odours, they spontaneously recognize individuals by their smell [10] and they prefer specific parts of human body for olfactory investigation [10,89], suggesting that different body parts produce specific odours that might convey different information.

Dogs discriminate conspecifics on the basis of their odour [90]. Moreover, they distinguish their own odour from that of others when presented with urine samples [90,91]. In social interactions, dogs engage in olfactory investigation as part of greeting behaviours to collect information about the other individuals [8]. They show a different interest for specific parts of conspecifics’ body for olfactory exploration, sniffing more intensely the face, the neck, the inguinal, and the perianal areas [8,10]. The odours are produced by different types of glands located in these areas, in particular, those located at the corner of the mouth, in the ear pinnae, the preputial and vaginal glands, and anal sacs. The particular interest shown in sniffing these areas suggests that dogs can obtain different information from the investigated regions, which may produce different odours [10]. Moreover, dogs collect social information by investigating other individuals’ urine and faeces placed in the environment [8]. Male and female dogs show a strong interest in unfamiliar urine and they investigate them to perform mate and threat assessment [92]. Olfactory communication includes scent marking behaviour, consisting of a first phase of investigation of other individuals’ marks followed by the deposit of the individual’s odours close to or on the existing marks [8,93]. Scent marking behaviour includes not only olfactory elements, but also visual and auditory components [11]. The placement of urine and faeces (the latter role has not been studied in detail) in the environment may be accompanied by ground-scratching behaviour, which adds both visual and auditory signals, produced by the act of scratching, and enrich the chemical signal of the mark with the deposition of interdigital glands secretions and with the dispersion of scats’ olfactory cues [8,11].

Communication via scents plays an important role in dogs’ reproductive behaviour. Bitches signal their reproductive status through urine marks and vaginal secretions [94], whose odour is extremely attractive for other dogs [95]. It elicits a specific reaction in male dogs, which deposit their own urine on or near to the females’ one as a signal for courtship [94].

Dogs can also release their odour in the environment by rolling on the ground, marking with their face and their entire body. Although this behaviour is still included in the canines’ repertoire and it maintains its communicative characteristic, it is no longer relevant for the evolutionary success of this species. Wolves use scent-rolling behaviour to pick up the scent of the pray and to carry it back to the pack, providing information about the health and location of the prey [8].

Moreover, it has been recently demonstrated that dogs are able to perceive the emotional content of conspecific odours, which induces behavioural and physiological effects in the receiver according to their valence [9,96]. Research specifically showed that during the sniffing of odours collected from perianal, interdigital, and salivary secretions soon after the end of a negative emotional event, in which the dog was left alone in an unfamiliar environment, dogs consistently used their right nostril. Given that the neural olfactory pathways ascend ipsilaterally to the brain, right nostril use reflects a main activation of the right hemisphere. In addition, the sniffing conspecific odours collected during “isolation” and “disturbance” situations causes an increase in heart rate and stress behaviours in dogs. Once again, these findings support the main role of the right side of the dog brain in the analysis of clear arousing signals.

Canines recognize humans by their odour. They are able to discriminate body odour of two identical twins living in the same environment [97] and to respond spontaneously to metabolic changes of their owner on the base of their scent [98]. Moreover, they associate the humans’ odour with previous experiences they had with them. They show an increase of their arousal state when presented with the veterinary sweat odour, which is generally associated with stressful experiences [95]; on the other hand, they associate familiar human odours with positive outcomes [99]. In a recent functional magnetic resonance imaging (fMRI) study, it is reported that familiar human odours activate the caudate nucleus, which is associated with positive expectations and reward, including social reward [99].

Dogs show a preference for investigating specific parts of the human body, and in particular, some specific areas of children’s bodies, namely the face and the upper limbs, suggesting that human odours produced at different anatomical parts could also provide different specific olfactory cues [9].

It has been recently found that, as for conspecific odours, dogs detect the emotional content of human odours, which induces different changes in their behaviour and in their cardiac activity [9,96]. In particular, an increase in behavioural and cardiac reactivity occurs during the sniffing of human odours collected during fearful situations. It is interesting to note that, contrary to that which has been observed for arousing conspecific odours, during the sniffing of the smell of human fear, bias in the use of the left nostril (i.e., left hemisphere activity) has been observed [9]. The latter suggest that chemosignals communicate conspecific and heterospecific emotions using different sensory pathways.

Moreover, dogs adjust their social behaviour toward humans according to the valence of the odour perceived, showing more stranger-directed behaviour when presented with the “happiness” odour, and more owner-directed behaviour when presented with the “fear” odour [96].

These findings demonstrate that chemosignals carry contextual-related information, supporting their specific role in dog–human communication.

## 5. Tactile Communication

Although rarely studied, tactile communication is an equally important aspect of dog communication. Tactile communication between dogs is used during agonistic interactions to impress an opponent (by an intense physical contact, putting paws over back or body of subordinate, grabbing the muzzle of the subordinate or young individuals and clasping another canids during ritualized aggression) or to maintain social bond (Figure 8) [8,100,101]. In particular, social cohesion is maintained by specific activities, such as resting in close contact (see Figure 9), placing the head over the shoulders of another dog during greetings or sexual approaches preceding mating, and by social grooming, which generally includes “face washing” (licking the other animal’s face) or “nibble” grooming, using the front teeth and rapid jaw open–close movements [8].

In a similar way, some human gestures during human–dog physical interaction could result in positive or negative canine emotional states, which drive to corresponding approaching and withdrawal behavioral responses of the dogs, even if they have been initiated with a different motivation [102]. People, equal if familiar or unfamiliar to a pet, tend to show their affection towards their pets by initiating physical contact. This is due to the fact that the tactile contact during human–dog interactions causes a series of benefits with regard to the physiology, the mental states, and the immune system of humans. For example, in humans, a decrease of both blood pressure and heart rate and an increase of the immune system function have been reported after petting dogs [103,104,105]. Otherwise, being petted serves as positive reinforcement for dogs as confirmed by associated heart-rate deceleration [106]. Nevertheless, physical contact in social interactions has different features in interspecific and intraspecific communication, in terms of both frequency and duration. Dogs rarely use physical contact to communicate with other individuals and tactile interactions (for example, grooming) are generally short lasting. On the contrary, humans tend to initiate and maintain physical contact with dogs with a higher frequency and longer duration, as it is a typical feature of human–human communication. For this reason, some dogs may appear less relaxed during human–dog tactile interaction (see Figure 10) tolerating physical contact or displaying a withdrawal behavioral response [107]. Some dogs tend to show discomfort using ambivalent signals and conflict behaviors during all close physical interactions and this phenomenon also depends on which specific part of their body is touched [108].

A significant influence of human-dog familiarity on dogs’ behavioral responses during tactile human-dog interactions has also been found. In particular, the work of Kuhne and colleagues [100] showed that dogs being petted by a familiar person showed significantly more appeasement gestures (e.g., blinking, looking elsewhere, closing both eyes, averted head, sitting, laying down, etc.), redirected behaviors (e.g., sniffing/licking on the floor, digging, drinking, visual scanning, etc.), and social approach behavior than dogs being petted by an unfamiliar person. Furthermore, significant differences in dogs’ behavioral responses depending on human-dog familiarity could be seen if the dogs were petted on specific parts of their body, supporting other findings that showed that dogs may generally dislike their hind legs, paws, and the top of their head being touched [109]. Dogs may interpret petting these specific canine body regions as agonistic communicative signals, which could create interferences with a normal and balanced human-dog bond [102]. Canines seem to better tolerate physical contacts (displaying less conflict and withdrawal behavioral responses) on the sides of their chest and under their chin. However, given that dogs’ reactions to handling depend on different factors (e.g., genetics and early experience, including socialization with humans, as well as physical and mental health, breed, learning and especially the context in which the interaction takes place [110,111]), there is no scientific concluding evidence yet concerning how to safely pet and play with dogs [109].

## 6. Conclusions

Dogs have a vast repertoire of visual, tactile, acoustic, and olfactory signals that they use for an expressive and fine-tuned communication with both conspecifics and humans. Nevertheless, the communicative importance of the different body parts in social interactions still remains poorly investigated. Future studies could evaluate dogs’ gaze pattern and olfactory attention toward human and conspecific bodies both in human–dog and conspecific interactions, in order to better identify which body regions are more informative for dogs during communicative interactions.

## Figures and Tables

**Figure 1 animals-08-00131-f001:**
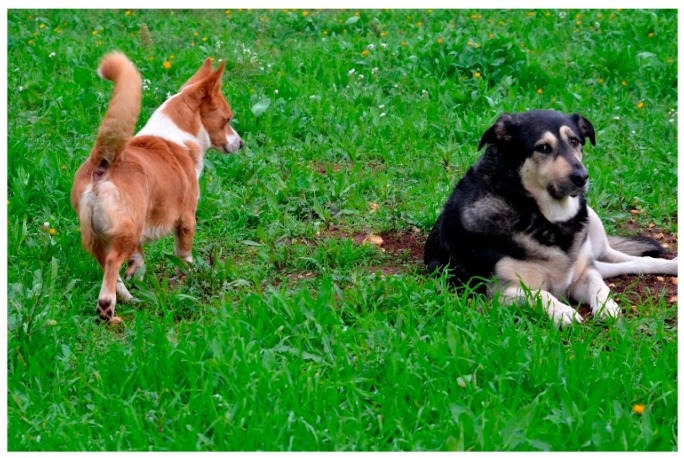
The female is looking at the little red male, asking him to increase the distance. The little red male is approaching in a curving but conflicting way; he has hackles and his face expresses tension. May be he is testing the reaction of the female, asking her to stand up; the female face expresses threat (she probably does not want to interact with him).

**Figure 2 animals-08-00131-f002:**
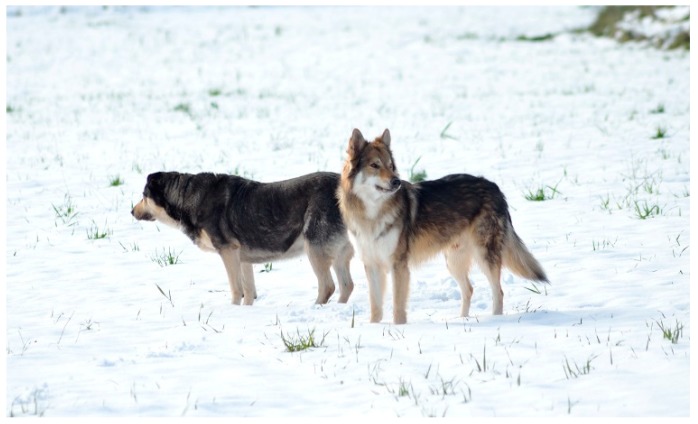
The two dogs have a very strong relationship. The Czech wolf needs to be close to his “adoptive mother” while he is looking at something that catches his attention. The female is looking at something else with a body language that gives information; she is much more self-confident.

**Figure 3 animals-08-00131-f003:**
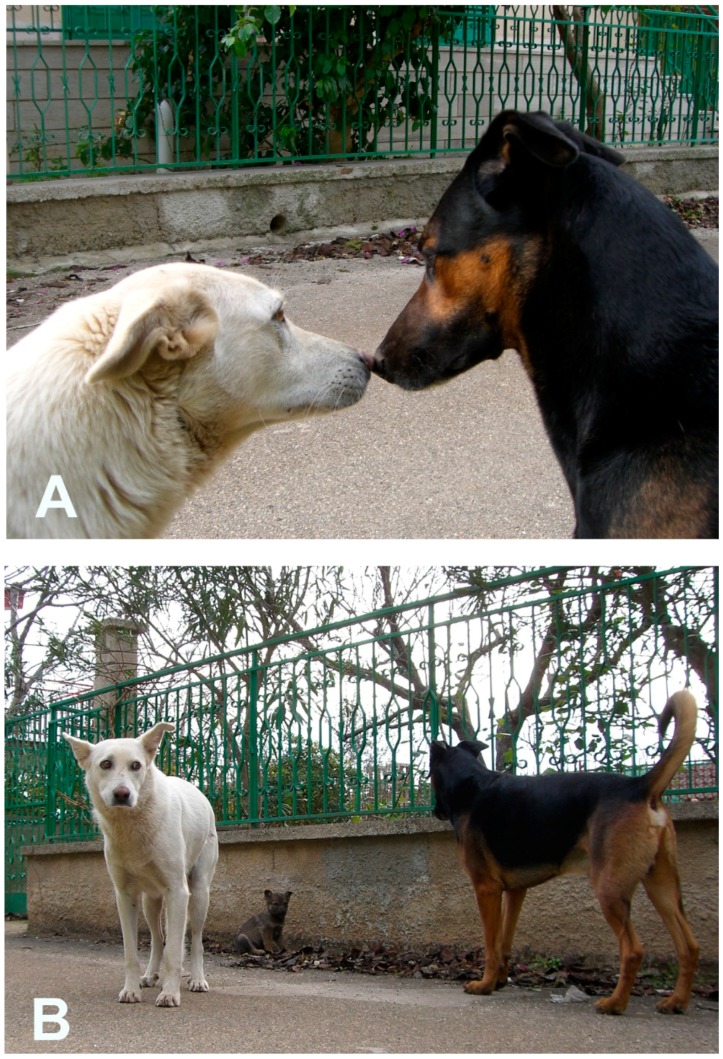
Free-ranging dogs. (**A**) The black male displays courtship behavior. His expression shows a closing distance request. (**A**,**B**) The female is showing her intention to avoid a conflict, but also her firm intention to enhance distance to protect her puppy.

**Figure 4 animals-08-00131-f004:**
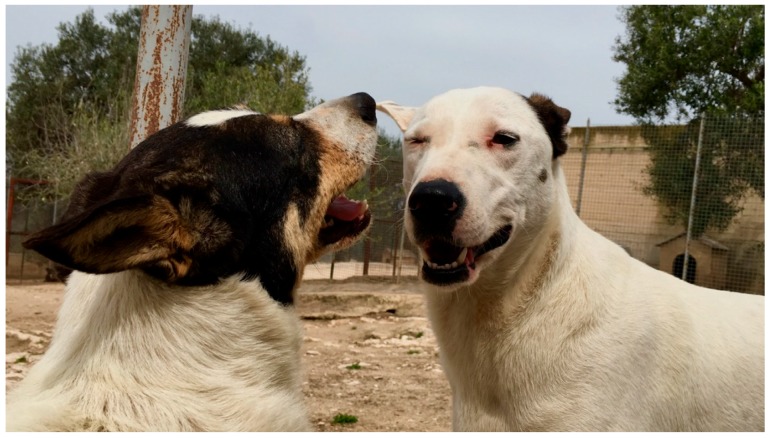
In this photo two, relaxed facial expressions are shown. The mouths are not tense, the looks are not direct, and the proximity tells us that the two dogs have a good relationship.

**Figure 5 animals-08-00131-f005:**
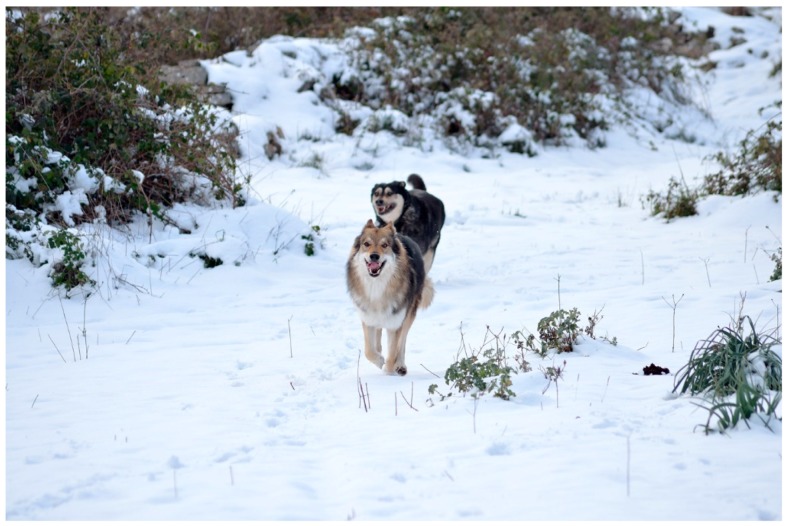
The Czech wolf is positively excited during play; his facial muscles are not in tension and his eyes are “soft”.

**Figure 6 animals-08-00131-f006:**
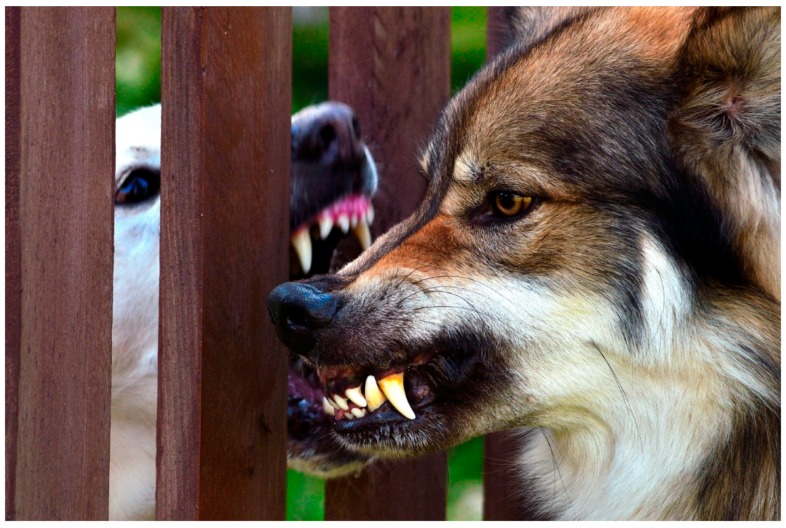
In this photo, the tension is very high: the Czech wolf is asking the other dog to back off, showing his desire to communicate; he is threatening the white dog, but his look is not directly at the other dog. The white dog instead is much more direct and intense (picture taken from a video footage).

**Figure 7 animals-08-00131-f007:**
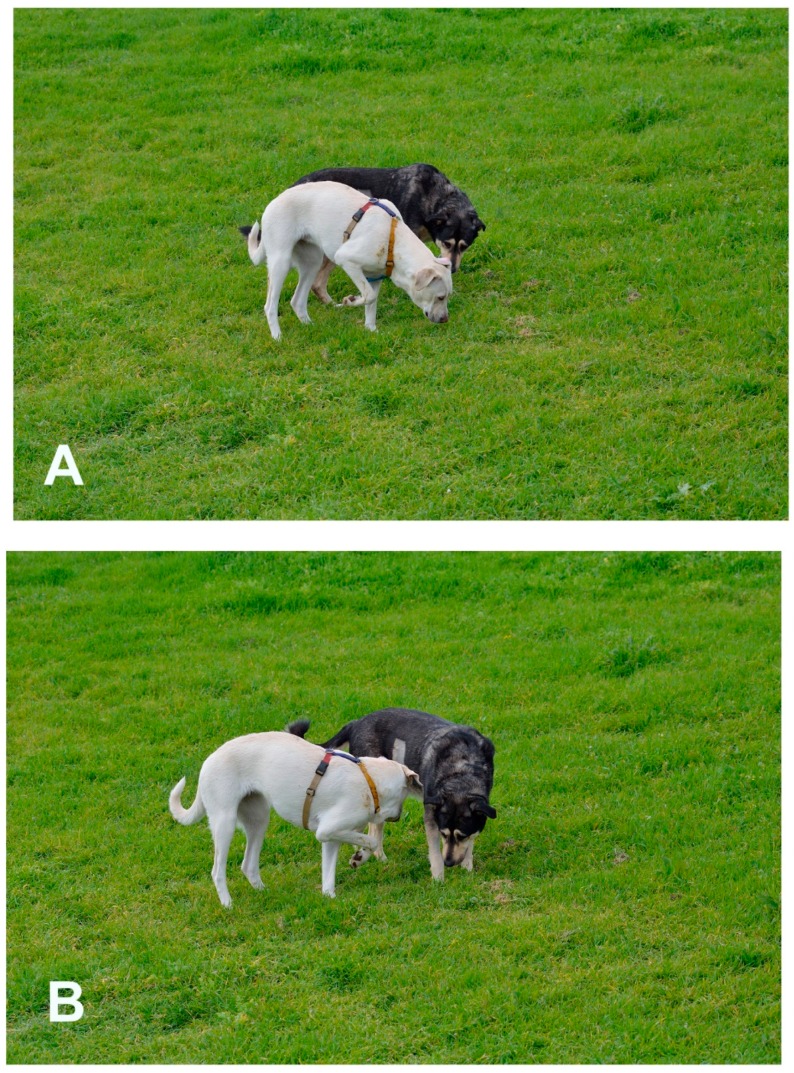
The white female is sniffing urine marking; the black female (with a shaved area on her right side due to an ecographic analysis) uses the urine marking as a resource to make clear a conflict with the white female. (**A**) In the first photo, the black female is asking distance and the eye contact is very clearly showing a threat. (**B**) In the second photo, the white female turns and goes away from the urine marking and the body language of the black female become more possessive; the direction of the head is on the urine marking, the direction of the eyes is on the white female (pictures taken from a video footage).

**Figure 8 animals-08-00131-f008:**
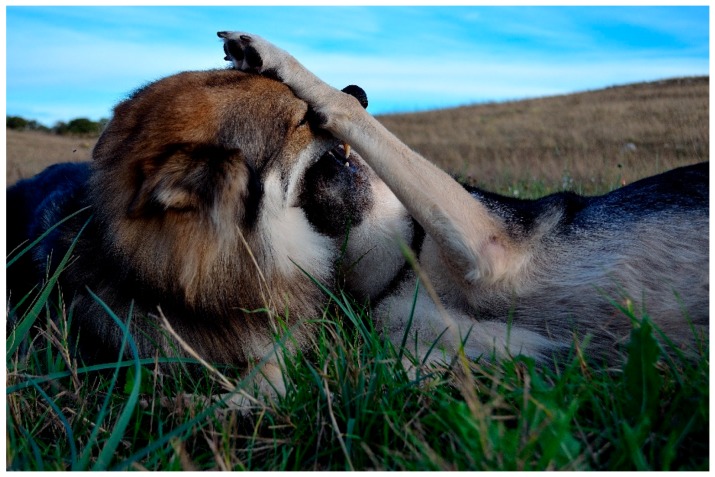
The Czech wolf is doing a muzzle grab during a bout of play.

**Figure 9 animals-08-00131-f009:**
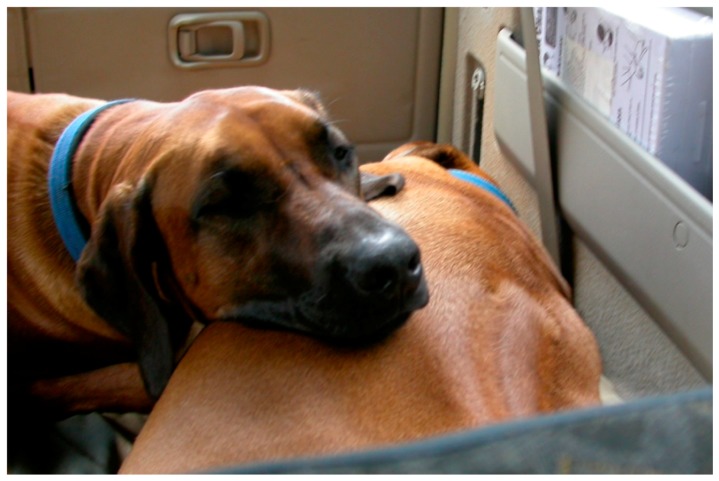
These two Rhodesian Ridgebacks usually sleep and rest in very close physical contact with each other. They have a very strong bond; the dog on the left is a daughter of the one on the right.

**Figure 10 animals-08-00131-f010:**
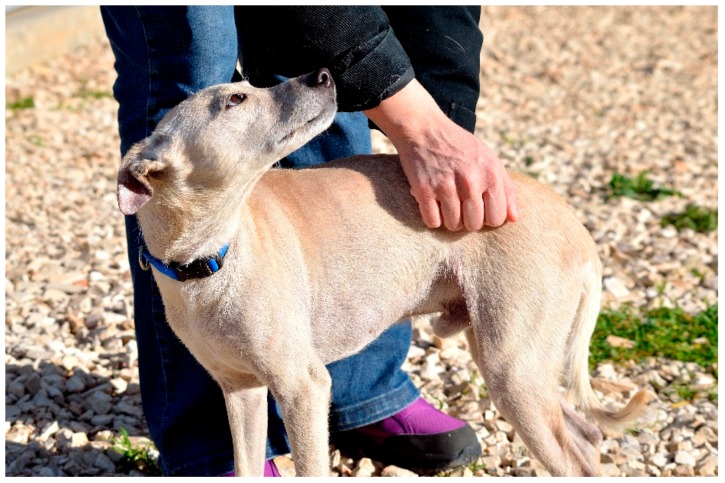
This dog is not relaxed during this tactile interaction. Although the physical contact is “gentle” (on the dog’s side and not on his head), the human is standing on the dog, making him feel uncomfortable.

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
