# Peer review of "Communication in Dogs"

_animals, 2018, doi:10.3390/ani8080131_

Round 1

Reviewer 1 Report

A very good and comprehensive review. I commend it for publication with Animals with the following suggestions:

Word choices:

Line 8 (and 29): "Communication takes place between members of the same species but it can also involve individuals belonging to different ones, as the long co-habitation process and inter-dependent relationship caused in domestic dogs and humans." Replace highlighted text with something like "...heterospecific individuals, as the..."

"dogs'" is used too frequently throughout the manuscript. Recommend replacing all occurrences in the main text with "canine" or "the canine". Some sentences will need minor adjustment to improve readability. For example line 189:

"The eyes informative role in human-dog communication is also demonstrated by dogs’ greater interest in investigating the eye region..."

as: "The informative role of the eyes for human-dog communication is also demonstrated by the greater interest by canids in the eye region ..."

Line 490: "Conflittuale" is not English - a better word choice might be 'hostile' in the given context.

Line 502: Replace "puppie" with puppy

Typographical errors:

Line 217: "owner’ signals than strangers’ ones" might be better stated as "...follow owner's signals rather than those from a stranger..."

Line 413: Moreover

Line 507: "(in fact they know each other)"

Author Response

All of the suggested revisions have been addressed (please see the red marked version of the manuscript).

Reviewer 2 Report

The present review is about dogs' communication with others (conspecifics as well as humans). The topic is very interesting and relevant, especially when considering the different or similar functions that one communicative signal can have according to the recipient's nature (other dog or human).

However, I have some comments:

- L 46: you mentioned dogs as involved in visual communication, as well as auditory and olfactory communication.

But why don't you mention tactile communication? Even if it is poorly documented, discussing its existence, and modalities, between dogs as well as between dogs and humans appear to me to be essential to your review.

For exemple, in Hubrecht, 1995 (The welfare of dogs in human care, in Serpell, The Domestic Dog: its Evolution, Behaviour and Interaction with people) it is reported that housing shelter dogs in pairs allow more tactile communication.

It is also important to discuss the fact that in dogs tactile communication is maybe not as important as in humans, and that however, humans still initiate a lot of tactile communication with dogs, that is sometimes too difficult for them to handle with (see for example: Kuhne et al., 2012, Effects of human-dog familiarity on dogs' behavioural responses to petting, Applied Animal Behaviour Science / or Kugne et al., 2012, Affective behavioural responses by dogs to tactile human-dogs interactions, Berliner und Münchener Tierärztliche Wochenschrift)

So I would strongly recommend to add a part on tactile communication, and I am sure such a part could enable you to add great figures.

- Also, you did not mentioned a recently described phenomenon in dog-human communication (also present in dog-dog communication of course): behavioural synchronisation and its role in promoting affiliation between individuals. Synchronising body position, body or head orientation, proximity (as in your Figure 2) and activity (as in your Figure 5) between dyadic partners is essential among individuals as it helps sharing inner states and maintaining social bonds between individuals.

I suggest that referring to at least those two review papers could help you discuss the field area, at least in your introduction:

Duranton & Gaunet, 2016 - Behavioural synchronisation from an ethological perspective (Adaptive Behavior) and Duranton & Gaunet, 2018 - Behavioural synchronisation and affiliation: dogs exhibit human-like skills (Learning & Behavior)

-Finally, my main concern is about the Figure' legends.

I think the Figures are really interesting and illustrate very well the review. However, the legends are sometimes not consistent with the text in their style: they are less scientific, less objective. In general, I would encourage you to give  the context more precisely (e.g. Figure 3: are they free ranging dogs? Figure 7: why one of the dog is shaved? did she had a recent surgery?). And also, I would encourage to replace words like "flirt" (Figure 3) or "territory" (Figure 6) that are not adequate in here as we do not know anything about the context of the figures.

To conclude, I really think this work deserve to be published, and encourage the authors to do the minor revisions ·  I am asking for, for the manuscript to gain clarity and be suitable for publication.

Author Response

However, I have some comments:

- L 46: you mentioned dogs as involved in visual communication, as well as auditory and olfactory communication.

But why don't you mention tactile communication? Even if it is poorly documented, discussing its existence, and modalities, between dogs as well as between dogs and humans appear to me to be essential to your review.

For exemple, in Hubrecht, 1995 (The welfare of dogs in human care, in Serpell, The Domestic Dog: its Evolution, Behaviour and Interaction with people) it is reported that housing shelter dogs in pairs allow more tactile communication.

It is also important to discuss the fact that in dogs tactile communication is maybe not as important as in humans, and that however, humans still initiate a lot of tactile communication with dogs, that is sometimes too difficult for them to handle with (see for example: Kuhne et al., 2012, Effects of human-dog familiarity on dogs' behavioural responses to petting, Applied Animal Behaviour Science / or Kugne et al., 2012, Affective behavioural responses by dogs to tactile human-dogs interactions, Berliner und Münchener Tierärztliche Wochenschrift)

So I would strongly recommend to add a part on tactile communication, and I am sure such a part could enable you to add great figures.

A new paragraph on canine tactile communication has been added. In addition three new figures have been uploaded. (please see the red-marked version of the manuscript). 

- Also, you did not mentioned a recently described phenomenon in dog-human communication (also present in dog-dog communication of course): behavioural synchronisation and its role in promoting affiliation between individuals. Synchronising body position, body or head orientation, proximity (as in your Figure 2) and activity (as in your Figure 5) between dyadic partners is essential among individuals as it helps sharing inner states and maintaining social bonds between individuals.

I suggest that referring to at least those two review papers could help you discuss the field area, at least in your introduction:

Duranton & Gaunet, 2016 - Behavioural synchronisation from an ethological perspective (Adaptive Behavior) and Duranton & Gaunet, 2018 - Behavioural synchronisation and affiliation: dogs exhibit human-like skills (Learning & Behavior)

A new paragraph discussing the suggested review papers has been added as follows (line 214):

"Recent studies demonstrated the existence of behavioural synchronization between dogs and humans (see for review [45]). The canine synchronizes its locomotor behaviour with that of its owner in different contexts, both indoors [46] and outdoors [47], and when facing an unfamiliar human. Dogs synchronize their behaviour with the owner’s withdrawal response toward strangers, taking longer time to approach them [48]. It has also been reported that the behavioural synchronization phenomenon is affected be dogs’ affiliation toward human: pet dogs show a higher performance in synchronizing their behaviour with their owner’ than shelter dogs with their caregivers. Moreover, behavioural synchronization affects dogs’ social preference toward humans, and in particular toward individuals synchronizing their locomotor activity with them [45]. Thus, authors concluded that, as previously described in humans, this phenomenon increases social cohesion and affiliation in dog-human dyads, contributing to emotional contagion [49]". 

Finally, my main concern is about the Figure' legends.

I think the Figures are really interesting and illustrate very well the review. However, the legends are sometimes not consistent with the text in their style: they are less scientific, less objective. In general, I would encourage you to give  the context more precisely (e.g. Figure 3: are they free ranging dogs? Figure 7: why one of the dog is shaved? did she had a recent surgery?). And also, I would encourage to replace words like "flirt" (Figure 3) or "territory" (Figure 6) that are not adequate in here as we do not know anything about the context of the figures.

Figure legends have been reviewed in their style (please see the red-marked version of the manuscript).

Reviewer 3 Report

Overall, I enjoyed reading this manuscript. The authors have done a good job in compiling references and summarising the state of the field of dog communication. I am particularly happy that the review will be open access.  

However, the review falls down in the use of English language and style. 

I have gone through the paper and highlighted the pdf where there are issues that need to be addressed. I included some suggestions, and also some questions for the authors in the comments. 

The Figure captions need some work in particular. The text that the authors have provided seems to over interpret the photograph, but perhaps these were taken from a video sequence where the extra information was available. This should be clarified. 

Please refer to the pdf for other questions and comments. 

Author Response

All the suggested comments have been addressed (please see the red-marked version of the manuscript).

Figure legends have been corrected.